# Experimental Determination of the Charge Carrier Transport Models for Improving the Simulation of the HR GaAs:Cr Detectors’ Response

**DOI:** 10.3390/s23156886

**Published:** 2023-08-03

**Authors:** Petr Smolyanskiy, Petr Burian, Mateusz Sitarz, Benedikt Bergmann

**Affiliations:** 1Institute of Experimental and Applied Physics, Czech Technical University in Prague, Husova 240/5, 110 00 Prague, Czech Republic; 2Faculty of Electrical Engineering, University of West Bohemia, Univerzitni 26, 108 00 Pilsen, Czech Republic; 3Danish Centre for Particle Therapy, Aarhus University Hospital, Palle Juul-Jensens Boulevard 99, 8200 Aarhus, Denmark

**Keywords:** hybrid detectors, particle tracking, X-ray detectors, models and simulations

## Abstract

The response of Timepix3 detectors with 300 µm and 500 µm thick HR GaAs:Cr sensors was studied with particle beams at the Danish Centre for Particle Therapy in Aarhus, Denmark. Therefore, the detectors were irradiated at different angles with protons of 240 MeV. The precise per-pixel time and energy measurements were exploited in order to determine the charge carrier transport properties. Using the tracks left by the penetrating charged particles hitting the sensor at the grazing angle, we were able to determine the charge collection efficiency, the charge carrier drift times across the sensor thickness, the dependency of the electron, and for the first time, the hole drift velocity on the electric field. Moreover, extracting the dependence of the charge cloud size on the interaction depth for different bias voltages, it was possible to determine the dependence of the diffusion coefficient on the applied bias voltage. A good agreement was found with the previously reported values for n-type GaAs. The measurements were conducted for different detector assemblies to estimate the systematic differences between them, and to generalize the results. The experimental findings were implemented into the Allpix Squared simulation framework and validated by a comparison of the measurement and simulation for the ${}^{241}$Am γ-ray source.

## 1. Introduction

The majority of hybrid pixel detector assemblies within the Medipix/Timepix family utilize silicon as the sensor material due to its well-established properties, low cost, and availability. However, recent advancements in crystal growth techniques have enabled the production of high resistivity chromium-compensated gallium arsenide (HR GaAs:Cr) [1], which presents an alternative to silicon, particularly in the detection and imaging of X- and γ-rays, where these sensors exhibit increased absorption efficiency.

In addition to their improved absorption efficiency, devices based on GaAs:Cr sensors also possess a high drift velocity of electrons (approximately six times that of electrons and more than ten times that of holes in silicon), making them suitable for utilizing the full timing precision of Timepix3 [2] (i.e., 1.6 ns) in applications such as particle tracking and for characterizing pulsed X-ray sources.

In order to accurately simulate various experimental configurations utilizing HR GaAs:Cr detectors, it is imperative to understand the transport properties of both types of charge carriers, despite the fact that non-collected charge carriers possess short lifetimes. Previous studies, such as [3,4], have demonstrated the impact of the holes on HR GaAs:Cr on the formation of pixel signals and the shape of energy spectra. The electron transport properties of HR GaAs:Cr have been thoroughly examined in previous investigations [5,6,7]. The transport properties of holes in HR GaAs:Cr have also been studied in [3,7] by utilizing two different methodologies. In our previous work [7], we investigated the hole’s mobility and lifetime through simulation/experiment comparisons of the hole’s drift time as a function of the interaction depth for a certain applied bias voltage. In the present study, we build upon this research by examining the dependence of the hole drift time and charge collection efficiency on the interaction depth for a range of bias voltages, thus allowing us to establish a proper dependence of the hole drift velocity on the bias voltage.

Charge sharing, a well-known phenomenon in detectors composed of small pixels, has been extensively studied in silicon-based Timepix detectors [8,9]. However, there is a lack of experimental studies on this effect in HR GaAs:Cr-based detectors [10]. In the present work, we utilize analysis of the characteristic elongated imprint left by energetic protons that penetrate the detector in the pixel matrix and study the dependence of the charge cloud profile width on the interaction depth at different bias voltages. This experimental finding was incorporated into a simulation tool, along with charge transport models, thus achieving accurate simulations of the detector response to γ-rays.

## 2. Instrumentation and Methods

### 2.1. Timepix3 Detectors with HR GaAs:Cr Sensors

The Timepix3 chip is a pixelated readout chip developed within the Medipix3 collaboration [11]. It features a 1.5625 ns time binning, data-driven readout, and simultaneous measurement of energy and time in each of its 256 × 256 pixels, with a 55 µm pitch. In the present work, we tested the HR GaAs:Cr-Timepix3 detectors with sensor thicknesses of 300 and 500 µm provided by Tomsk State University. While both rely on ohmic contacts, the 300 µm thick HR GaAs:Cr sensor’s backside is made of gold, and the 500 µm thick sensor has a nickel backside contact.

The per-pixel energy calibration was conducted with X-ray fluorescences and characteristic γ-rays as described in [12]. In order to suppress the influence of the charge losses, in the hole collection (positive polarity), the sensor was irradiated from the chip side.

The lowest noise-free energy threshold was found to be 4.7 keV for the 500 µm thick detector and 3.9 keV for the 300 µm thick detector. For detector control, read out, and sensor biasing, the Katherine interface [13] was used. The temperature of the sensors and chips during the operation was around 50 °C.

The majority of measurements and simulations have considered only the signal of electrons created in the sensor, due to their considerably larger lifetime (lifetime $\tau_{e}$ > 10 ns [1,3,5,7]) compared to holes $\tau_{h}$, which have been determined for different samples and by different methods to be 0.2 ns [14], 1.4 ns [3], and 4.5 ns [7]). Literature values of electron and hole mobilities in GaAs are $\mu_{e}$ = 3200 – 4700 cm^2^/V/s [1,5] and $\mu_{h}$ = 210 – 320 cm^2^/V/s [1,7], respectively.

### 2.2. Data Preprocessing

Data were sent off the chip in a stream of pixel hits. After applying energy calibration and time-walk correction [15], the acquired data were sorted chronologically and split into pixel sets with a maximum time difference of 200 ns. Within these groups, spatially adjacent pixels were put together into a “cluster”, which was then characterized by the following features. The *cluster volume* is its total energy content, calculated by summing up the energies $E_{i}$ detected in the pixels: $E_{\mathrm{cluster}}=\sum_{i}^{N}E_{i}$, with *N* being the number of pixels in the track, also referred to as the *cluster size*; the *cluster height* is the maximum energy measured in a single pixel within the cluster: $E_{\mathrm{height}}=\max\left\{E_{0},\cdots E_{N}\right\}$; $t_{\min}$ is the lowest measured time within a single pixel of the track since measurement start; $x_{\mathrm{centroid}}$, $y_{\mathrm{centroid}}$ denotes the coordinates of the energy-weighted geometrical center of a cluster.

### 2.3. Mobility–Lifetime Product Measurement Principles

For measurement of the mobility–lifetime product of the electrons $\mu_{e}\tau_{e}$, the well-known approach based on the Hecht relation was applied. The relation was only modified for the case of small pixels [16]. The detectors were illuminated from the common electrode side by fluorescence photons from Zr foil ($K_{\alpha_{1}}$ = 15.775 keV) while the bias voltage was swapped in the ranges of [−350; −35] V and [−300; −30] V for the 300 µm and 500 µm thick detectors, respectively. Photons of this energy have an average range in GaAs of less than 25 µm. Therefore, the majority of interactions occurred close to the backside contact, and the charge transport properties could be investigated almost for the full thickness of the sensor.

In order to measure the average value of the $\mu_{e}\tau_{e}$-product (across the sensor), the energy spectra recorded by each pixel were summed and then analyzed at different bias voltages (see Figure A1). The Zr photopeaks were fitted by a Gaussian, and the obtained means $E_{\mathrm{meas}}^{\mathrm{Zr}}$ were plotted against the bias voltage $U_{\mathrm{bias}}$. The dependencies of $E_{\mathrm{meas}}^{\mathrm{Zr}}$ versus $U_{\mathrm{bias}}$ for each pixel then were fitted by the function from [16]:(1)$$Q\left(U_{\mathrm{bias}}\right)=Q_{0}\times \left[\frac{d}{\mu_{e}\tau_{e}\times \left(U_{\mathrm{bias}}-U_{\mathrm{th}}\right)}\times \int_{0}^{d}\phi \left(z\right)\times e^{-\frac{-zd}{\mu_{e}\tau_{e}\times \left(U_{\mathrm{bias}}-U_{\mathrm{th}}\right)}}dz+e^{-\frac{d^{2}}{\mu_{e}\tau_{e}\times \left(U_{\mathrm{bias}}-U_{\mathrm{th}}\right)}}\right],$$
where $Q_{0}$ is the charge generated by the single photon, *d* is the detector thickness, $U_{\mathrm{th}}$ is the bias voltage above which the induced charge become above the given detector threshold, and ϕ(z) is the weighting potential. It was calculated numerically using the Allpix Squared framework [17].

### 2.4. Proton Beam Measurement and Data Analysis

The Timepix3 detectors were irradiated with 240 MeV protons. The detectors were positioned in the beam at angles of 75 and 85 degrees with respect to the sensor normal. The bias voltage of the detectors was scanned in the ranges of [−300; −25] V, [+40; +300] V and [−500; −25] V, [+55; +500] V for the 300 µm and 500 µm thick detectors, respectively. In order to have the signals in the pixels caused by holes or electrons, the Timepix3 chips were configured either in the electron or hole collection modes [7].

To clean the measured data sets, we further used only clusters that had a *cluster height*, *cluster volume*, and *cluster size* within 3σ of the mean values. Clusters with edge pixels were rejected to avoid wrong energy and time measurements.

Figure 1 shows the detector response to the 240 MeV protons impacting at an angle of 75 degrees with respect to the sensor normal in the form of imprints in the pixel screen indicating the energy left by the ionizing particle, for electron and hole collection. Figure A2 shows the energy deposition spectra of the 240 MeV protons registered by a 500 µm thick detector for different bias voltages.

In hole collection, the short lifetime significantly reduces the range of the holes, so that the holes created close to the backside are not able to reach the region, where they would be able to induce enough charge to trigger a pixel. Thus, the tracks seen in the positive polarity (hole collection) are shorter than the tracks for the electron collection. This fact was used in further analysis. Moreover, the slower collection of the holes leads to an increased width of the tracks.

Even though in the hole collection only part of the particle trajectory is seen, since the impact angle θ is known, by determining the exit point $x_{\mathrm{exit}}$, the interaction depth along the track can be calculated as:(2)$$z\left(x\right)=|x-x_{\mathrm{exit}}|\times \;\tan\left(\theta \right).$$

We hereby define z=0 to be the pixel plane. The drift time $t_{\mathrm{drift}}$ at pixel *x* is given by the difference of the pixel timestamp $t_{x}$ and $t_{\min}$. With Equation (Equation 2), these drift times can then be related to the interaction depth, finding $z(t_{\mathrm{drift}})$. The slopes of the drift time’s dependence on the interaction depths at different bias voltages were fitted with a line, so that the drift velocity $V_{\mathrm{drift}}$ could be studied as a function of the electric field strength *E*. In the same way, by utilizing the per-pixel energy deposition information, the dependence of the charge collection efficiency (CCE) on the interaction depth *z* was obtained.

### 2.5. Simulation in the Allpix Squared Framework

The Allpix Squared [17] is an open-source framework for the Monte-Carlo simulation of silicon detectors. Beginning from version 2.3 it supports additional sensor materials: gallium arsenide, germanium, cadmium telluride, cadmium zinc telluride, diamond, and (4H) silicon carbide. For the HR GaAs:Cr, the Ruch–Kino model [18] of the electron’s drift velocity dependence on the electric field was applied. It has been found to be a good description for the charge carrier motion in the GaAs:Cr sensor [5]. In this work, we confirmed that this model worked well for different HR GaAs:Cr sensors (see Section 3.2.1). In the current version of Allpix Squared, the hole mobility is supposed to be configured manually in the configuration file, even though the hole drift velocity model has been determined (see Section 3.2.2). The lifetimes of the holes and electrons can be also adjusted from the configuration file. The influence of induction on the drift time measurement (see the discussion in [19]) was accounted for by using the *TransientPropagation* module of the Allpix Squared.

## 3. Experimental and Simulated Results

### 3.1. Mobility–Lifetime Product of Electrons

The evaluation methodology for the mobility–lifetime product determination was described in Section 2.3. The results in the form of $E_{\mathrm{meas}}^{\mathrm{Zr}}$ dependence on the applied bias voltage are shown in Figure 2 for the 300 and 500 µm thick detectors. The obtained values of the mobility–lifetime product values for electrons $\mu_{e}\tau_{e}$ are $\left(4.09\pm 0.25\right)\times 10^{-4}$ cm^2^/V and $\left(1.24\pm 0.02\right)\times 10^{-4}$ cm^2^/V for the 300 µm and the 500 µm thick detectors, respectively. These values are consistent with previous studies [4,5,6].

### 3.2. Drift Velocity Models

#### 3.2.1. Electrons

It has been shown in previous works [5,7] that the Ruch–Kino model, which describes the dependence of the electron drift velocity $V_{\mathrm{drift}}^{e}$ on the electric field *E*, works well for the HR GaAs:Cr material:(3)$$V_{\mathrm{drift}}^{e}=\left\{\begin{array}{c} \mu_{e}\times E,\;\mathrm{if}\;E\leq E_{0}\\ \mu_{e}\times E/\sqrt{1+{\left(E-E_{0}\right)}^{2}/E_{c}^{2}},\;\mathrm{if}\;E>E_{0} \end{array},\right.$$
where $\mu_{e}$ is the low field mobility of an electron, $E_{0}$ is the critical field below which the drift velocity linearly increases with an increasing electric field, and the field $E_{c}$ determines the saturation velocity.

However, in previous work the acquired data points were not sufficient for the precise determination of the Ruch–Kino model parameters. Therefore, taking the previous findings into account, we increased the number of data points in the bias range most sensitive to the parameters $E_{C}$ and $E_{0}$. During the measurements, the sensor temperatures were stabilized at ∼ 50 °C. Figure 3 shows the measured dependencies of the electron drift velocity on the electric field for the two studied detectors. The experimental points were fitted with the function (Equation 3), and the low field mobilities of the electron were determined as $\mu_{e}$ = (1887 ± 53) cm^2^/V/s and $\mu_{e}$ = (5858 ± 119) cm^2^/V/s for the 300 µm and the 500 µm thick detectors, respectively. As one can see, the shapes of the fitted curves were similar for the detectors of different thicknesses, but since they were from different batches (which can lead to different characteristics of the crystals) it is not possible to draw any conclusions about the generalization of the results.

As seen in Figure 3, it is concluded that HR GaAs:Cr-based detectors should not operate at maximum bias voltage to provide the best timing performance. Using a bias voltage of −180 V, the drift time across the 500 µm thick GaAs:Cr sensor was found to be 4 ns, which was 10 times lower than in silicon with the same thickness [15].

Combining the latter results with the findings from Section 3.1, we calculated the lifetimes of the electrons to be (22 ± 1) ns and (21 ± 1) ns for the 300 µm thick and 500 µm thick detectors, respectively.

#### 3.2.2. Holes

The hole drift velocity was determined in the same way as the electron drift velocity. The results for the two detectors are presented in Figure 4. At the electric field of 10 kV/cm, the hole drift velocity was close to saturation at values of ($1.62÷1.95)\;\times \;10^{6}$ cm/s for both detectors. The results were consistent with the previously published results [20]; only the saturation electric field was lower in our case.

The hole lifetime determination was conducted by comparing simulations performed with different sensor parameters with the measured data. The hole lifetimes in the range of (1.3 ÷ 1.7) ns and (4.5 ÷ 5.5) ns were used as the simulation input parameters for the 300 and 500 µm thick sensors, respectively. As seen in Figure 5, the best agreement of the experiment and the simulation was found for $\tau_{h}$ = (1.5 ± 0.2) ns (300 µm) and $\tau_{h}$ = (5.0 ± 0.5) ns (500 µm).

### 3.3. CCE Dependencies on Interaction Depth

Figure 6 shows the dependence of the charge collection efficiency for both the investigated detectors when used in electron collection. For the 300 µm thick detector, it is presented for the first time, while the results from the 500 µm sensor were in agreement with previous works [5,6] and are shown for different bias voltages for completeness.

The short lifetime of the holes led to a poor CCE in the hole collection mode (see Figure 7). Throughout the set of investigated bias voltages, the charge carriers generated close to the common electrode were trapped before they the could reach the pixel electrodes.

### 3.4. Charge Cloud Size vs. Interaction Depth

When an electric field is applied to the sensor, the charge carriers created by the ionization move from the generation point to the cathodes and anodes and induce a charge in the closest lateral pixel and sometimes in its neighboring pixels. During this drift motion, the charge carriers undergo lateral diffusion and repulsion, resulting in the broadening of the charge cloud. Collectively, these effects are referred to as charge sharing. For energy depositions below 100 keV, repulsion is not considered, since it is not significant. The solution to the diffusion equation for an initial point-like distribution is a Gaussian distribution. The dispersion of this distribution can be characterized as [21]:(4)$$\sigma =\sqrt{2Dt_{\mathrm{drift}}}=\sqrt{2D\times \frac{z}{V_{\mathrm{drift}}}},$$
where *D* is the diffusion coefficient of the charge carriers, and $V_{\mathrm{drift}}$ is the drift velocity of charge carriers.

Analyzing Figure 3, one can expect that since the drift velocity of the electron, after saturation, started to decrease, the *cluster size* would follow the opposite behavior (since electrons are slower, the lateral diffusion is larger, following from (Equation 4). But, a simple experiment showed the reverse result. A 500 µm thick detector was placed 15 cm from the ${}^{241}$Am source, and the bias scan was performed in the range from −30 V to −500 V. During the analysis, only clusters from the photopeak were selected, and their mean size was calculated for the different bias voltages. The resulting plot is shown in Figure 8. The *cluster size* decreased with the increase in the bias voltage, which may be caused by the dependence of the diffusion coefficient on the bias voltage. Since, as shown in Section 3.2.1, the electron mobility is a function of the bias voltage, the diffusion coefficient is also a function of the bias voltage:(5)$$D\left(U_{\mathrm{bias}}\right)=\frac{\mu_{e}\left(U_{\mathrm{bias}}\right)k_{B}T}{e},$$
where $k_{B}$ is the Boltzmann constant, *T* is the sensor temperature, *e* is the elementary charge.

In order to investigate the impact of charge sharing and to determine the diffusion coefficient dependence on the bias voltage, we applied the approach adapted from [22]. The method relies on detecting the lateral expansion of the track created by a minimum ionizing particle as it passes through the sensor at a grazing angle. In order to analyze the data, the dataset from Section 2.4 was utilized, which pertains to an impact angle of 75 degrees with respect to the normal of the sensor. The tracks were filtered based on the following specific attributes: the energy $E_{\mathrm{cluster}}\in \left[1900,2800\right]$ keV, the size $N_{\mathrm{pixels}}\in \left[40,65\right]$ pixels, and the length L∈[28,36] pixels.

To obtain accurate measurements, the method involved the removal of the lateral pixels triggered by δ-electrons and replacing the high energy values in certain pixels with the median value over the track. The resulting set of pixels was used to calculate the energy-weighted average coordinates, which were then fitted with a linear function to determine the position along the particle trajectory with subpixel resolution (see Figure 9). Further analysis was limited to the tracks tilted at ϕ = 0.5–1.5 degrees with respect to the *x*-axis (azimuth angle), as they provided the most precise fit.

The determination of the depth coordinate is explained in detail in Section 2.4. For each depth bin, the energy deposition was calculated by fitting the corresponding spectrum with a convolution of a Landau curve and a Gaussian. Then, for each *x* coordinate (which corresponds to the *z* coordinate) of the track, the deviation between the detected *y* position and the position $y_{\mathrm{subpixel}}$ determined by fitting was calculated as $\Delta y_{z}=y-y_{\mathrm{subpixel}}$. By combining the measured energy information (or induced charge) $MPV_{z}$ in each bin *z* with the corresponding position $\Delta y_{z}$, the lateral charge deposition profile of the pixel was created for different interaction depths. These dependencies can be fitted using the function described in [22]:(6)$$MPV_{z}∼\frac{MPV_{0,z}}{2}\times \left[1+\mathrm{Erf}\left(\frac{p/2-\Delta y_{z}}{\sqrt{2}\sigma \left(z\right)}\right)\right],$$
where $MPV_{0,z}$ is the most probable value of the Landau fluctuations of the energy deposition, *p* = 55 µm is the pixel pitch, and σ(z) describes the Gaussian spread of the transverse profile of the charge carrier cloud.

Figure 10a,b display the lateral charge deposition profiles for the depth bins z=100 µm and z=400 µm, respectively. They were fitted using function (Equation 6). The resulting values of σ(z) are: σ(z=100 µm) = (6.7 ± 0.4) µm and σ(z = 400 µm) = (16.8 ± 0.9) µm. It can be observed that the width of the flat region was larger when close to the pixels compared to when close to the common electrode, indicating a higher contribution of charge sharing. Furthermore, the amplitude of the dependencies differed, which could be attributed to the loss of charge during the drift of the charge carriers from the interaction point towards the pixels, as shown in Section 3.3.

Figure 11a illustrates the relationship between the width of the charge cloud profile, denoted by σ(z), and the interaction depth *z* measured at the bias voltage of −300 V. To extract the value of the diffusion coefficient *D*, the experimental points were fitted with the function (Equation 4). Notably, when the interaction happens in proximity to the common electrode, the induced charge is more likely to be shared among adjacent pixels. It is important to mention that the results are influenced by several major factors, including the deposited energy (as the initial size of the charge cloud is energy-dependent) and the bias voltage (as it affects the lateral diffusion).

Having conducted the measurements at various bias voltages, it was possible to obtain the dependence of the diffusion coefficient *D* on the electric field *E*. The result is shown in Figure 11b. D(E=0) was calculated through (Equation 4) for $\mu_{e}$ = 5858 cm^2^/V/s. The obtained results were consistent with the pioneer investigations of the Gunn effect in n-type GaAs [18,23].

### 3.5. Simulation Verification with X-rays

Additional verification of the electron/hole transport models was performed by comparing the simulated and measured response of the 500 µm thick GaAs:Cr-Timepix3 detector to the γ-rays of 59.5 keV. The measured values of the mobilities and lifetimes were used in the configuration file of the Allpix Squared framework. The determined diffusion coefficient dependence on the bias voltage was incorporated in the *TransientPropagation* module. A custom digitization module from the Medipix collaboration was used.

To accumulate the experimental data, the detector was irradiated with an ${}^{241}$Am source placed at a distance of 15 cm from the sensor to minimize the influence of other low energy lines of the ${}^{241}$Am source. Figure 12a compares the measured and simulated energy spectra in electron collection mode, showing an overall very satisfactory agreement. The small discrepancy around 50 keV can be explained by the differing CCE across the area of the GaAs:Cr sensors [24], which was not taken into account in the simulation.

The correct simulation of the charge sharing effect is proven by the comparison of the experimental and simulated *cluster size* distributions (see Figure 12b), which agree within 5%.

## 4. Conclusions

The response of the Timepix3 detectors with HR GaAs:Cr sensors layers of different thicknesses (300 and 500 µm) was thoroughly investigated using a laboratory γ-ray source, as well as energetic proton beams. By leveraging the disparity in the track lengths between the hole collection and electron collection, we successfully determined the lifetimes of the holes, yielding values of (1.5 ± 0.2) ns and (5.0 ± 0.5) ns for the 300 µm and 500 µm thick sensors, respectively. Additionally, employing an innovative analysis approach, we measured the dependence of the hole drift velocity on the electric field and the diffusion coefficient’s dependency on the electric field in HR GaAs:Cr. These experimental findings were then incorporated into the simulation framework Allpix Squared, resulting in a commendable agreement between the measured and simulated data.

## Figures and Tables

**Figure 1 sensors-23-06886-f001:**
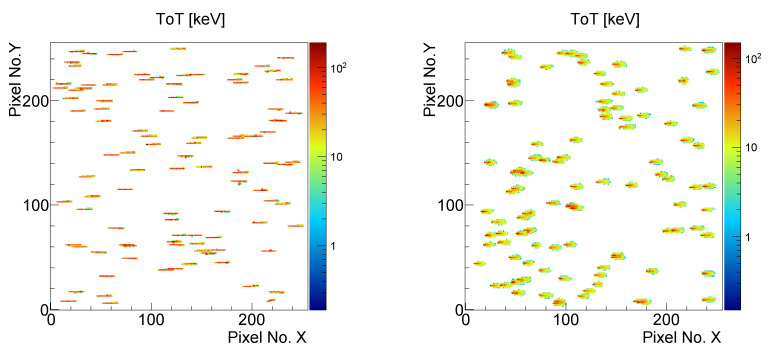
Per-pixel energy deposition for typical particle tracks measured by the 300 µm thick detector in a 240 MeV proton beam at an impact angle of 75 degrees for the electron (**left**) and the hole (**right**) collection modes. The short lifetime of the holes together limits their drift distances; so the shorter tracks are observed in positive polarity, while the slower collection leads to increased lateral width.

**Figure 2 sensors-23-06886-f002:**
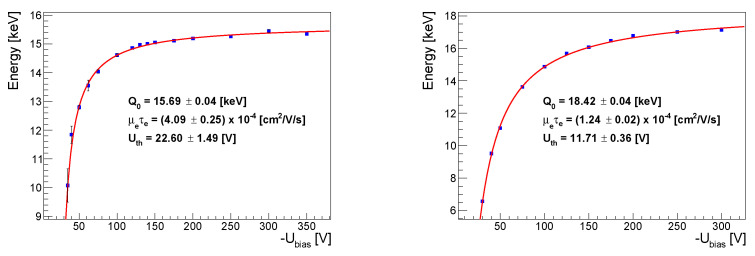
The position of the Zr photopeak in the dependence on the applied bias voltage for the 300 µm (**left**) and for the 500 µm (**right**) thick detectors. The $\mu_{e}\tau_{e}$ values were determined by fitting with function (Equation 1).

**Figure 3 sensors-23-06886-f003:**
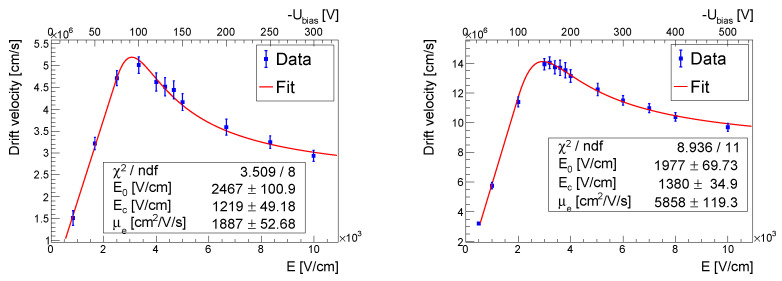
Electron drift velocity as a function of the electric field for the 300 µm (**left**) and for the 500 µm (**right**) thick detectors.

**Figure 4 sensors-23-06886-f004:**
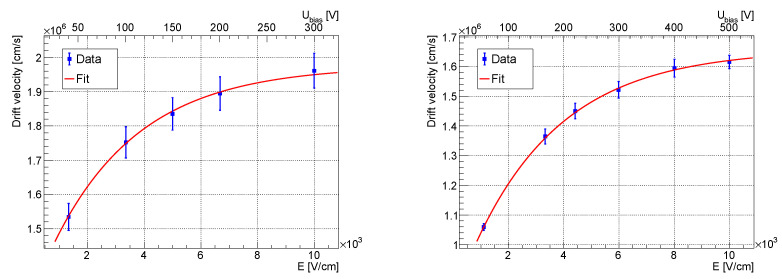
Hole drift velocity as a function of the electric field for the 300 µm (**left**) and for the 500 µm (**right**) thick detectors.

**Figure 5 sensors-23-06886-f005:**
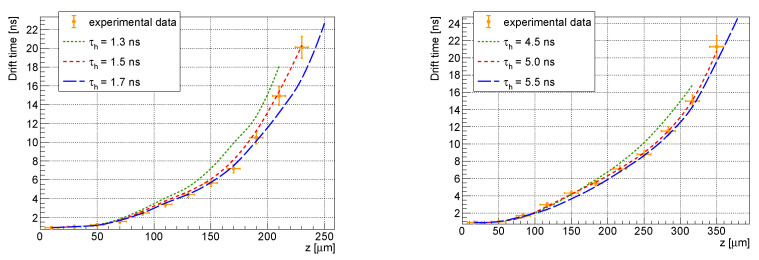
Simulated and experimental dependencies of the drift time on the *z* coordinate for various hole lifetimes $\tau_{h}$ for the 300 µm (**left**) and for the 500 µm (**right**) thick detectors for the bias voltage of 300 V. The red dashed line indicates the best fit to the data points, while the blue dashed and green dashed lines indicate the upper and lower bounds of the lifetime, respectively.

**Figure 6 sensors-23-06886-f006:**
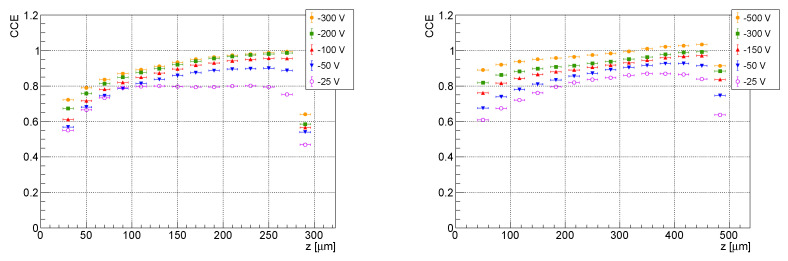
Experimental dependencies of the charge collection efficiency of electrons on the interaction depth *z* for the 300 µm (**left**) and the 500 µm (**right**) thick detectors.

**Figure 7 sensors-23-06886-f007:**
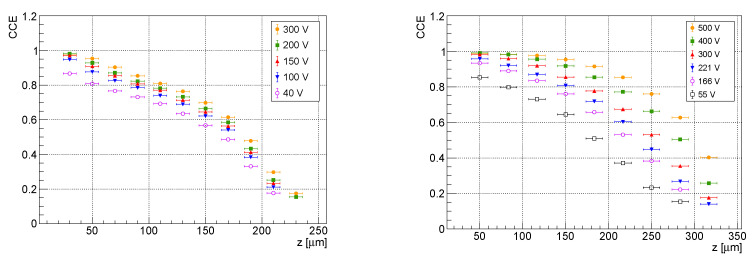
Experimental dependencies of the charge collection efficiency of the holes on the interaction depth *z* for the 300 µm (**left**) and for the 500 µm (**right**) thick detectors.

**Figure 8 sensors-23-06886-f008:**
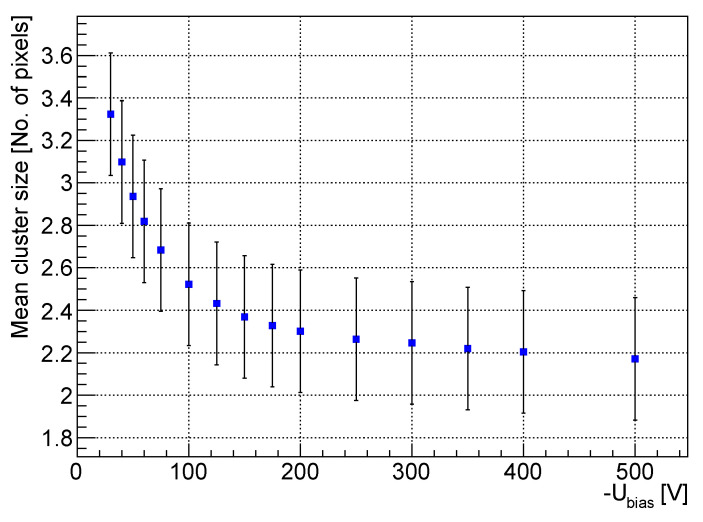
The mean size of clusters formed by the 60 keV photons from the ${}^{241}$Am source in the dependence on the bias voltage (for the 500 µm thick detector).

**Figure 9 sensors-23-06886-f009:**
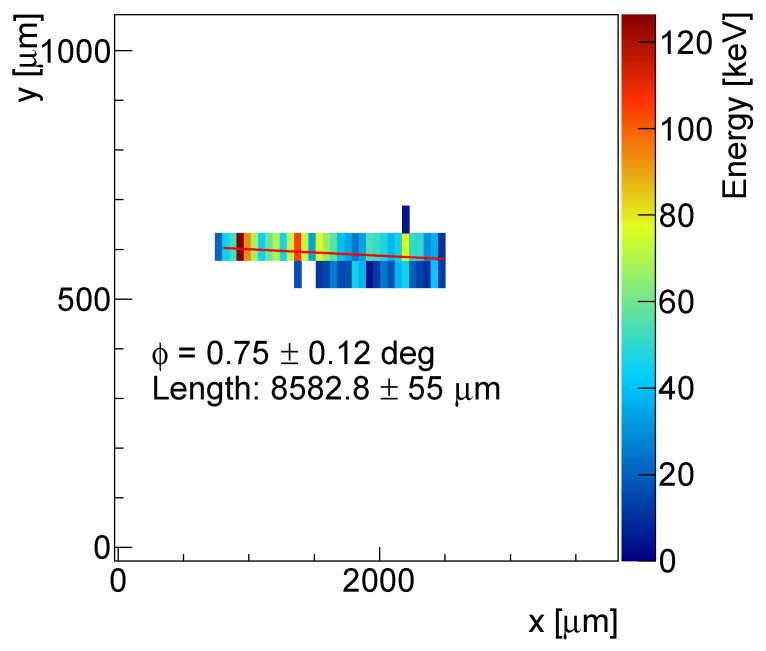
Example of a proton track fitted with a linear function.

**Figure 10 sensors-23-06886-f010:**
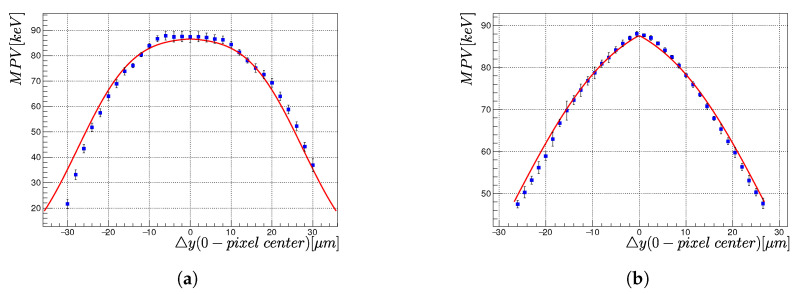
The lateral charge deposition profile measured at depth *z* = 100 µm (**a**) and *z* = 400 µm (**b**) for the bias voltage of −300 V.

**Figure 11 sensors-23-06886-f011:**
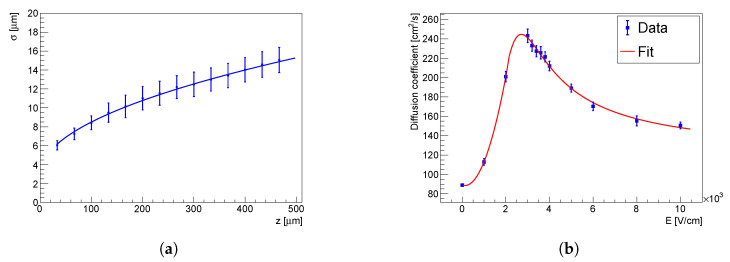
The dependence of the charge cloud profile width σ(z) on the interaction depth *z* for a bias voltage of −300V (**a**). The dependence of the diffusion coefficient on the electric field measured for the 500 µm detector (**b**).

**Figure 12 sensors-23-06886-f012:**
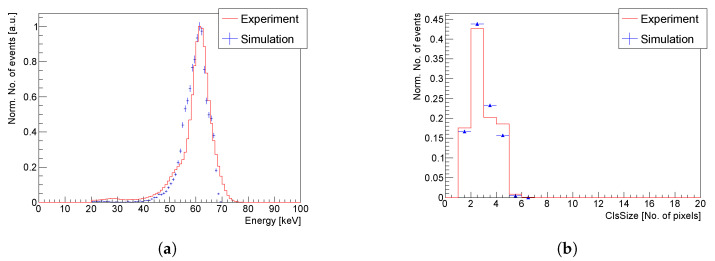
Simulated and experimental energy spectra of the ${}^{241}$Am source (**a**). Comparison of the experimental and simulated distributions of the *cluster size* for irradiation by the ${}^{241}$Am source (**b**).

## Data Availability

Not applicable.

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
