# Peer review of "Experimental Determination of the Charge Carrier Transport Models for Improving the Simulation of the HR GaAs:Cr Detectors’ Response"

_sensors, 2023, doi:10.3390/s23156886_

Round 1

Reviewer 1 Report

This paper deals with the experimental determination of some properties of the charge carriers in HR GaAs:Cr detectors.  These parameters are, at least partially, unknown, and accurate measurements are useful for a deep understanding of transport processes in these detectors. The topic is relevant, because accurate measurements of these transport parameters do not exist, at least to my knowledge, and this research fills that gap. I feel that the measurements techniques used are adequate, and sufficienctly accurante.  I do not think of possible other checks to be done.

The paper is very well written, clear, easy to read, and contains all the relevant details and info about the experimental procedure and the analyis of the results obtained. I advised it to be published as it is.

Author Response

Dear Reviewer,

thanks a lot for your feedback!

Reviewer 2 Report

The paper "Experimental determination of charge carrier transport models for improving simulation of the HR GaAs:Cr detectors response" shows a nice study on the charge carrier of GaAs:Cr. 

The paper describes quite in detail a well conducted study, also supported by simulation. We think that the quality is enough for publication. Overall, the quality of the plots should be improved a bit. Some plots look as master thesis plots  (small legends, writings on top of each other). It would make the paper more scientific sounding. 

- In reference 4, it is shown that the pixel is not really square i.e. there is a large variation pixel by pixel. Did you consider a large number of pixels, such to obtain an 'average' response or did you use only the same pixels in your measurements, been biased to the response of a particular pixel?

-line 63, can you specify why you chose such a high temperature. Is it the temperature at the electronics or of the sensor? If it is normal in such operations and if you expect the findings to be highly modified using an ambient cooled environment (sensor at 30 degrees roughly). 

- line 76[-35, -350] V->  please, use the more common notation in which the range   is expressed with the lower value first, and the higher values afterwards -> [-350, -35]. This is needed in several places along the text, substitute everywhere.

-line 221: please specify in words (whiteout mathematical notation) what the quantities and the ranges are that is difficult to follow

- line 227: did I understand correctly that you accepted from the previous 75 degrees tilt to the current 1-3 degrees tilt?

-lines 256-260: Am I right right to say that this effect could be studied also with X-rays of different energies which are absorbed at different depth (or tilting the sensor with a beam of X-rays). 

- All section 3.5. It is not clear if you changed some parameter from the default values of the simulation: in case you did, specify which parameter and how it was changed. In case you did not changed them , it would be nice to redo the simulation with parameters changed closer to what you find and see if the data/simulation distribution agree better.

- no need to define CCE on line 6

- line 20 define HR there

-line 56, small  'S -> 's

-line 77 line -> energy

-line 189: remove 'Due to'

Author Response

Dear Reviewer,

thanks a lot for your feedback! We've used your comments to improve the paper. All comments are in the attached file.

Reviewer 3 Report

The paper is well organised and written. No relevant issues with English language have been noticed (only a very few typos). The subject is of sound interest in the field and this is reflected in the abstract. There is a noticeable tension between the abstract, the introduction, the overall development of the subject and the concluding remarks, where the highlights of the work are somewhat understated.

The level of self-citation is at ~30%, and partly justified by the reference to a long standing activity about the subject.

The experimental part is well presented and the data analysis supported by a detailed recourse to method, literature, equations and plots, thus facilitating the reader through the subject development.

A few more comments on details are in order, on which the Authors' feedback would be appreciated:

Lines 80-84

The Authors discuss about energy spectra and how they extract the average E_meas from Zr photopeak, would it be useful for the reader to display a sample spectrum? This would apply also to Am source and proton beam as well to give an idea of how the "raw data" look.

Eq. (1):

The Authors represent Q = Q0 * f(mu, tau, etc.), but I would invide them to check the algebraic rearrangement, since in Ref. [14] the equation shows the charge ratio Q/Q0, therefore I would expect Q0 should also be factor of the rightmost exponential (brackets missing?).

Figure 3 and related text discussion:

The curves and experimental data show a very precise and "universal" shape for both plots, is there any conclusion or ansatz about the scaling with thickness which could be mentioned or referred to?

Figure 3 and Figure 11:

The drift velocity and the diffusion coefficient display a very similar shape. Is there any underlying reason for that, according to the Authors' experience (I may be missing the point...)?

The abscissa units are [V/cm] in Fig. 3, [kV/cm] in Fig. 11. Am I missing something, or this factor 1000 may be a typo?

Fig. 12:

The comparison of data with simulation in the leftmost plot (Am source calibration) is briefly discussed in the paper, but I may have missed any comment on the width of the distributions, which appears to be significantly different. Can the Authors comment on that?

References:

Can the Authors review the references? For instance, Ref. 7 does not display the journal. Ref. 16 has the authors names italicised. 

Other typos along the text:

Line 56: sensor'S backside

Line 95: Data send off -> Data sent off

Line 100: energies detecting -> energies detected

Line 101: The cluster height -> the cluster height

Line 102: max E0,...,EN -> max{E0,...,EN}

Line 134: confirmed -> confirm

Line 176-178: the Authors should mention both 300 and 500 um sensors result, it seems only one is reported ... please check

Line 242: There were fitted using -> They were fitted using

In conclusion, the overall level of the article is certainly deserving publication, conditionally to replying to the comments and performing a more extensive check for typos.  

English language is at a good level, only a few typos here and there. Please check (see also previous comments).

Author Response

(The authors gave the same response as above.)
